# Impact of Prematurity on Auditory Processing in Children

**Maria Y. Boboshko** [1,2,*], **Irina V. Savenko** [2], **Ekaterina S. Garbaruk** [2,3], **Veronika M. Knyazeva** [1] and **Marina J. Vasilyeva** [1,*]

[1] Department of Higher Nervous Activity and Psychophysiology, St. Petersburg State University, 199034 St. Petersburg, Russia; v.m.knyazeva@spbu.ru

[2] Laboratory of Hearing and Speech, Pavlov First St. Petersburg State Medical University, 197022 St. Petersburg, Russia; irina@savenko.su (I.V.S.); kgarbaruk@mail.ru (E.S.G.)

[3] Scientific Research Center, St. Petersburg State Pediatric Medical University, 194100 St. Petersburg, Russia

[*] Correspondence: boboshkom@gmail.com (M.Y.B.); marinamarinajv@gmail.com (M.J.V.)

**Abstract:** Prematurity is one of the most crucial risk factors negatively affecting the maturation of the auditory system. Children born preterm demonstrate high rates of hearing impairments. Auditory processing difficulties in preterm children might be a result of disturbances in the central auditory system development and/or sensory deprivation due to peripheral hearing loss. To investigate auditory processing in preterm children, we utilized a set of psychoacoustic tests to assess temporal processing and speech intelligibility. A total of 241 children aged 6–11 years old (136 born preterm and 105 healthy full-term children forming the control group) were assessed. The preterm children were divided into three groups based on their peripheral hearing status: 74 normal hearing (NH group); 30 children with bilateral permanent sensorineural hearing loss (SNHL group) and 32 children with bilateral auditory neuropathy spectrum disorder (ANSD group). The results showed significantly worse performance in all tests in premature children compared with full-term children. NH and SNHL groups showed significant age-related improvement in speech recognition thresholds in noise that might signify a "bottom-up" auditory processing maturation effect. Overall, all premature children had signs of auditory processing disorders of varying degrees. Analyzing and understanding the auditory processing specificity in preterm children can positively contribute to the more effective implementation of rehabilitation programs.

**Keywords:** preterm children; auditory processing disorders; sensorineural hearing loss; auditory neuropathy spectrum disorder; temporal processing; gap detection test; speech intelligibility; Russian matrix sentence test

## 1. Introduction

Babies born alive before 37 completed weeks of pregnancy are considered preterm. Depending on gestational age, there are extremely preterm (less than 28 weeks), very preterm (28 to less than 32 weeks), and moderate to late preterm (32 to 37 weeks) babies. The WHO estimates that approximately 1 in 10 babies are born prematurely, and in 2020, 13.4 million babies were born preterm [1]. The achievements of modern peri- and neonatology contribute to increasing numbers of premature babies who survive, but despite this, the frequency of developmental disorders, including sensorineural anomalies, in preterm children remains high and may even increase as they grow older [2,3]. All this requires careful monitoring of the development of premature babies, especially from the extremely preterm group, to minimize the negative consequences of prematurity.

Children born preterm even without severe neurodevelopmental and neurosensory disabilities can exhibit poorer speech and language development, deterioration of higher mental function formation, reading skills, academic achievement, and social–emotional development than term-born children, not only in early childhood but also in the long term [4–7]. The occurrence of these disorders is primarily due to a violation of the normal

morphofunctional development of all sensory systems of a premature baby, and the auditory system is no exception. This is due, firstly, to the immaturity of the hearing organ at birth, early exposure to the extrauterine environment, as well as the high vulnerability of immature auditory structures to damaging factors [8–11]. Apart from prematurity itself, a number of other factors have been associated with neurodevelopmental impairment including peripheral hearing loss and auditory processing disorders (APDs) in preterm children. These include perinatal and neonatal hypoxia–ischemia, extended stays in the neonatal intensive care unit (NICU), neonatal hyperbilirubinemia requiring exchange transfusion, a longer period of assisted ventilation and prolonged respiratory support, acquired hypoxic–ischemic encephalopathy, and others [12–16].The coexistence of many perinatal adverse factors can negatively affect the development of the auditory system of a premature baby with the formation of insufficiency at any level. At the same time, the dysfunction of the central auditory nervous system (CANS) can manifest at any age. Recent research shows that preschool-age children who were born very preterm demonstrate sensory processing deficits in several domains including auditory [11,17–19].

Auditory processing of any acoustic information, including speech signals, requires active and continuous interplay between peripheral and central parts of the auditory system providing perception, recognition, analysis, and storage of the incoming acoustic signal. The normal functioning of CANS provides the ability of localization and lateralization of sound stimuli, differentiation of sounds, recognition of acoustic signals, analysis of the temporal characteristics of acoustic information, perception of reduced sound information, as well as sounds in the presence of a competing acoustic stimulus [20]. Morphologically in ontogenesis, CANS is formed during the 7–8th week of gestation [21], and the auditory system of the fetus becomes functional at around 25 weeks of gestation [22], when it starts reacting to pure tones [23], thus providing the possibility to register the auditory brainstem response (ABR), which indicates the functioning of peripheral sensory input [24,25]. The period from 25 weeks of gestation to 6 months of life was believed for a long time to be "critical" for the normal development of the child's auditory system, when the processes of its maturation under the influence of acoustic experience are the most active and when any damaging factor can disrupt its development [22,26]. However, as it has now become known, cortical structures are influenced by external factors even before the onset of the critical period. In early corticogenesis, there is a period of spontaneous activity of transient neural networks that function autonomously even before the full maturation of the sensory (including auditory) periphery takes place. The subplate neurons (SPNs) play a leading role in the functioning of these networks. The zones of SPNs were earlier considered to be a "waiting zone" for the development of the thalamocortical projections. Currently, SPNs have been determined to be actively involved in the regulation of endogenous spontaneous activity of neural networks, which precedes activity-dependent development, playing an integrative/instructive role in early cortical development. The role of the SPNs is active and irreplaceable in the formation of global thalamic–cortical–thalamic networks, cortical columnar structures, maturation of intracortical inhibitory connections, etc. [27]. The peak of the SPN zone development occurs in the midgestation period (between 24 and 32 weeks) when the transient cortical network is most vulnerable. Early pathological conditions, such as hypoxia, inflammation, or exposure to pharmacological compounds, alter spontaneous activity patterns, which subsequently induce disorders in cortical network activity [27,28].

Thus, any damaging factor affecting the developing sensory system in a certain period of time can further contribute to disorders of auditory processing in premature children and lead to delays or disorders in the CANS maturation process [8,29]. These abnormalities can persist for a long time and, even in the absence of peripheral auditory impairment, lead to central auditory processing deficits that may affect different levels of the auditory system and be present at different ages. Symptoms may become apparent in the preschool and early school years or at a later academic stage of the child's life due to changes in the acoustic environment or increased academic demands [30].

It is well known that children born preterm are at a high risk of developing peripheral hearing disorders. Hearing loss may be primarily caused by damage to the hair cells, resulting in sensorineural hearing loss (SNHL), or damage at the level of the inner hair cells, synapse, or auditory nerve, leading to auditory neuropathy spectrum disorder (ANSD) [14,16,31].

Information regarding auditory processing in children born preterm is rather scarce, and it is practically absent in relation to premature children with peripheral hearing loss. Continuously weakened auditory input is assumed to negatively affect the subsequent auditory processing ability. Several studies using psychoacoustic methods showed signs of CAPD in preterm children, which primarily manifested as deficits in temporal processing [32–35], impaired dichotic listening [35], and speech perception under conditions of competing stimuli (in noise) [34,35]. A high percentage of children with peripheral hearing impairment were also shown to struggle with acoustic information processing due to sensory deprivation [36–39]. These children may have difficulties with auditory temporal ordering and temporal resolution skills [40] and speech perception in noisy environments [41–43].

Psychoacoustic testing is the most effective, informative, and reliable method to evaluate auditory processing in children and it can be used from 4 years of age [20,44,45]. Normative data for behavioral measures of central auditory functioning for children younger than 4–5 years old are often limited or not available due to task complexity and maturational variability of the CANS. Many electrophysiologic measures of central auditory function yield variable results in children younger than 10 years of age. The use of assessment procedures requires a thorough understanding of the effects of maturation on the test results [46]. Auditory processing test performance is assumed to be an index of auditory system development, speech perception, and auditory discrimination. To achieve this, a combination of non-speech and speech tests is used, which enables the elimination of the influence of cognitive processes on test results and differentiation between central auditory dysfunction and impaired processing of linguistic information [20,47].

The effectiveness of central auditory processing results from the morphofunctional state of the related brain structures, and the child's CANS has a high potential for plasticity, not only in the "sensitive" period, which is limited to 2–3 years of life but also at a later age, which can positively influence processes in the CANS [48,49].

The aim of this study was to assess the age-related dynamics of central auditory processing in preterm children, both with normal hearing and with peripheral hearing loss.

We hypothesized that (1) central auditory processing is impaired in premature children with both normal peripheral hearing and hearing loss when using hearing devices, and (2) the deficits in auditory processing are gradually compensated for due to the neuroplasticity as the child becomes older.

## 2. Materials and Methods

### 2.1. Participants

A total of 241 children aged 6–11 years old participated in the study. Among them, 136 children born preterm were assessed, with 132 individuals born at 32 weeks of gestation or less and four children born at 33–35 weeks of gestation. The preterm children were divided into three groups based on their peripheral hearing status. Group 1 consisted of 74 preterm children (mean gestational age of 28.5 ± 2.3 weeks; mean birthweight of 1218 ± 427 g) with normal peripheral hearing function (NH group). Group 2 included 30 preterm children (mean gestational age of 28.6 ± 1.7 weeks; mean birthweight of 1105 ± 220 g) with a bilateral permanent sensorineural hearing loss (SNHL group) from mild to moderately severe degree. Group 3 included 32 preterm children (mean gestational age of 28.9 ± 2.4 weeks; mean birthweight of 1291 ± 561 g) with bilateral auditory neuropathy spectrum disorder (ANSD group) from mild to severe degree. The results revealed no significant difference in the gestational age and birthweight between the preterm groups. The control group (Group 4) included 105 healthy full-term children (mean gestational age

of 38.1 ± 0.9 weeks; mean birthweight of 3215 ± 570 g) with normal hearing. In the control group, no known perinatal risk factors for hearing loss and deafness were identified, and there were no indications of recurrent middle ear diseases. Within each group, three age subgroups were identified: 6–7 (defined as "a"); 8–9 (defined as "b") and 10–11-year-old children (defined as "c") (see Table 1).

**Table 1.** Groups and subgroups of preterm and full-term children.

|  | Group 1: (NH) | Group 2: (SNHL) | Group 3: (ANSD) | Group 4: (Control) | Total |
|---|---|---|---|---|---|
| 6–7 years old, subgroup (a) | 22 | 10 | 8 | 50 | 90 |
| 8–9 years old, subgroup (b) | 25 | 10 | 10 | 29 | 74 |
| 10–11 years old, subgroup (c) | 27 | 10 | 14 | 26 | 77 |
| Total | 74 | 30 | 32 | 105 | 241 |

Groups of preterm and full-term children: group 1 (NH)—preterm children with normal hearing; group 2 (SNHL)—preterm children with sensorineural hearing loss; group 3 (ANSD)—preterm children with auditory neuropathy spectrum disorder; group 4 (Control)—full-term children with normal hearing.

### 2.2. Audiological Assessment of Peripheral Hearing

All participants underwent a comprehensive audiological assessment that included the basic audiological examination and central auditory processing testing, along with routine otolaryngologic examination and a detailed study of medical history. During the basic audiological examination, impedancemetry (tympanometry and acoustic reflexometry with ipsi- and contralateral stimulation), pure tone audiometry (PTA), registration of otoacoustic emissions (OAEs), and auditory brainstem response (ABR) recording (with a cochlear microphonic test, if necessary) were performed.

Peripheral hearing function was considered within the normal range when the pure tone thresholds did not exceed 15 dB nHL in the standard frequency range; a tympanogram was type "A"; ipsilateral and contralateral acoustic reflexes were present and OAEs and ABR were within normal range.

Audiological signs of SNHL were the following: normal tympanogram, OAEs absence, and elevated ABR thresholds consistent with pure tone thresholds.

ANSD was diagnosed if ABRs were absent or markedly abnormal, OAEs and/or cochlear microphonic potential were present, and a tympanogram was type "A". Hearing loss degree in children with ANSD was determined based on pure tone thresholds.

The degree of hearing loss was classified according to ASHA recommendations [50] based on the average air conduction thresholds at four frequencies (mean PTA0.5, 1, 2, 4): normal, ≤25 dB; mild, 26–40 dB; moderate, 41–55 dB; moderately severe, 56–70 dB and severe, 71–90 dB.

The age range for SNHL diagnosis in the 2nd group was 2 months to 3 years, with a median age of 6 months (3 months of corrected age). The age range for ANSD diagnosis in the third group was 3 months to 6 years, with a median age of 4.8 months (2.5 months of corrected age). The first hearing aid fitting was performed in children aged from 6 months to 6 years. Hearing-impaired children used binaural hearing aids (59 children) or cochlear implants (three children received a unilateral cochlear implant at the age of 2.4, 2.6, and 2.7 years). Aided pure tone thresholds assessed in the free sound field at frequencies of 0.5 kHz, 1 kHz, 2 kHz, and 4 kHz in all children were within 30–40 dB.

In this study, 57 children who had just been identified as having hearing loss were enrolled in an early intervention program. All of the children received regular sessions with a speech therapist, and 25 of them continued to receive therapy at the time of the research. One of the inclusion criteria was a sufficient level of speech, which was necessary for speech testing. Speech and language assessments were conducted by a speech therapist for all of the children, and they all demonstrated proficiency in extended sentence speech.

The study was conducted in accordance with the Declaration of Helsinki, and approved by the Ethics Committee of St. Petersburg Psychological Society (protocol #21,

6 April 2023). Informed consent was obtained from all the subjects/subjects' parents involved in the study.

### 2.3. Assessment of Central Auditory Processing

To evaluate the central auditory processing, a set of psychoacoustic tests was administered, available for use from the age of 5. We used the Random Gap Detection Test (RGDT) and the Duration Pattern Sequence (DPS) Test to examine temporal processing (temporal resolution and temporal ordering). The Russian matrix sentence test (RuMatrix) and its simplified version (Simplified RuMatrix) in quiet and noise conditions were used to assess speech intelligibility. For children 6–9 years old, the Simplified RuMatrix test was implemented, and for children 10–11 years old, we used the RuMatrix test. All audiological procedures were performed in the acoustically shielded room. For normal-hearing children, testing was carried out using headphones; for hearing-impaired children with hearing aids/cochlear implants, all testing procedures were performed in a free sound field with one loudspeaker located at 1 m in front of the child. The following equipment was used for pure tone audiometry, speech audiometry, and non-verbal psychoacoustic tests: clinical audiometer Maico MA42 (Berlin, Germany), headphones TDH39, loudspeaker SVEN SPS-700 (Kotka, Finland), portable CD player AEG portable mp3, CD with non-verbal tests, a laptop with Oldenburg Measurement Application software v. 1.3 (HörTech GmbH, Oldenburg, Germany), EarBox sound card (Auritec, Hamburg, Germany), and Sennheiser HDA200 headphones with a loudspeaker from Genelec (Berlin, Germany) were used for the RuMatrix test. Statistical analysis was performed using an independent *t*-test. The significance was considered at $p < 0.05$.

### 2.3.1. Random Gap Detection Test

To evaluate the auditory temporal resolution, the Random Gap Detection Test (RGDT) was administered [51]. During the training phase, participants were provided with a training block to ensure they understood the task correctly. During the testing phase, pure tones at frequencies of 0.5, 1, 2, and 4 kHz were presented at a comfortable sound level and separated by silent intervals. Each acoustic signal had a duration of 15 ms. Signals with silent gaps ranging from 0 to 40 ms were presented in random order. A total of 9 signals at each frequency were used. The task of the listener was to answer if the presented signal was perceived as one sound or two. The minimum silent interval at which the listener perceived two sounds was measured. The gap detection threshold was calculated as the average of the thresholds obtained at the tested frequencies. Typically, gap detection thresholds at the three frequencies should be less than 20 ms. In cases in which the gap detection threshold exceeded 40 ms for two or more frequencies, the test was considered to have failed [51].

### 2.3.2. Duration Pattern Sequence Test

The Duration Pattern Sequence (DPS) test was used to examine the processing of auditory temporal ordering. An acoustic signal that consisted of the frequency spectrum equivalent to three different musical instruments—oboe, violin, and piano—was used. Each trial comprised three signals with long (600 ms, L) and short (300 ms, S) durations and with a 300 ms pause between signals. The signals were combined into five patterns with different durations: SLS, SSL, LLS, LSL, and SLL. A total of 15 trials were presented to every participant, with each possible pattern repeated three times. The children were required to provide a nonverbal response by pointing at the appropriate picture placed in front of them. The pictures contained combinations of short and long stripes indicating the duration of the pattern. The rate of correct rhythm recognition was assessed and reported as a percentage [52–54].

### 2.3.3. Russian Matrix Sentence Test

To assess the speech intelligibility in 10–11-year-old children, the Russian matrix sentence test (RuMatrix) and its simplified version (Simplified RuMatrix) for children

6–7 and 7–8 years old were used. The RuMatrix test comprised five-word semantically unpredictable sentences consisting of a name, a verb, a numeral, an adjective, and a noun, such as "Peter sees ten old books", while the Simplified RuMatrix speech material test utilized three-word sentences composed of a numeral, an adjective, and a noun, such as "five red halls". During testing, the participant had to repeat the sentence or at least the individual words. The test included 20 sentences for the Russian matrix sentence test and 14 sentences for the simplified version presented in quiet and noise with an adaptive procedure. In the first phase, the speech recognition threshold in quiet ($SRT_Q$), measured in dB SPL, was evaluated. In the second phase, the children were tested in the presence of background noise. A fixed noise level of 65 dB SPL was used. Two tracks of 14 or 20 sentences were presented, with the first track considered as the training phase. The intensity of the speech signal was automatically adjusted, decreasing when the listener answered correctly and increasing in case of an incorrect response. The speech recognition threshold in noise ($SRT_N$) measured in dB SNR was estimated. All measurements were conducted in an open-set format, where subjects repeated sentences they heard without any visual support [55,56].

### 2.4. Statistical Analysis

The data were analyzed using Statistica software (v.10, StatSoft, Inc., Tulsa, OK, USA). The distribution of the data was checked using Shapiro–Wilks W tests. Student's *t*-tests were used for variables that had normal distributions, while the nonparametric Mann–Whitney test was used for those that did not when comparing the groups. As a criterion of significance, a 95% confidence level ($p < 0.05$) was chosen.

## 3. Results

### 3.1. The Peripheral Hearing Assessment

Children in the first group and the control group had normal peripheral hearing according to the results of the basic audiological examination. In the second group, bilateral permanent SNHL was observed in all children. The degrees of hearing loss are presented in Table 2. Two children with mild hearing loss exhibited a ski-slope hearing loss pattern characterized by increased thresholds at high frequencies. All children were bilateral hearing aid users. The results of the basic audiological examination in the third group confirmed bilateral ANSD in 32 children with varying degrees of hearing loss (Table 2). A total of 21 children had no ABRs (thresholds of more than 100 dB nHL) and 11 children had markedly abnormal ABRs with ABR thresholds of 80 dB or more. OAEs disappeared in all children with age; the determination of a cochlear microphonic potential during previous examinations confirmed the presence of ANSD. A total of 29 children in this group used hearing aids and three children were cochlear implant users.

**Table 2.** Distribution of hearing loss degree in children with SNHL and ANSD.

| Degree of Hearing Loss | SNHL Group $n = 30$ | ANSD Group $n = 32$ |
|---|---|---|
| Mild to Moderate (Me = 50.5 dB HL; range: 35–54.8 dB HL) * | 27 | 14 |
| Moderately severe (Me = 60.7 dB HL; range: 56.3–70 dB HL) | 3 | 14 |
| Severe (75 dB HL) | 0 | 1 |
| Profound (unilateral CI: 91.2; 92.5; 93.8 HL) ** | 0 | 3 |

* mean air conduction PTA at four frequencies (0.5; 1; 2; 4 kHz). ** mean air conduction PTA for unaided ear.

### 3.2. The Central Auditory Processing Assessment

3.2.1. Random Gap Detection Test

The Random Gap Detection Test (RGDT) performance was classified into three types: (1) the test was successfully performed and test results were within the normative values (average gap detection threshold at three frequencies was less than or equal to 20 ms); (2) the test was successfully performed, but an average gap detection threshold exceeded the normative values (ranged from 20 to 40 ms) and (3) the test was not completed (average gap detection threshold exceeded 40 ms). The RGDT results are presented in Figure 1. All full-term children from the control group passed the test completely. In the NH group of preterm children, 6.8% did not pass the test, 67.5% had the average gap detection threshold ranging from 20 to 40 ms, and 25.7% of children successfully performed the test with normal gap detection thresholds. Among hearing-impaired children, only 6.7% of children with SNHL and 9.4% of children with ANSD successfully performed the RGDT and demonstrated normal gap detection thresholds. A total of 33.3% of the hearing-impaired children with SNHL and 37.5% with ANSD passed the test but their average gap detection threshold exceeded the normative values; finally, 60.0% with SNHL and 53.1% with ANSD did not complete the test.

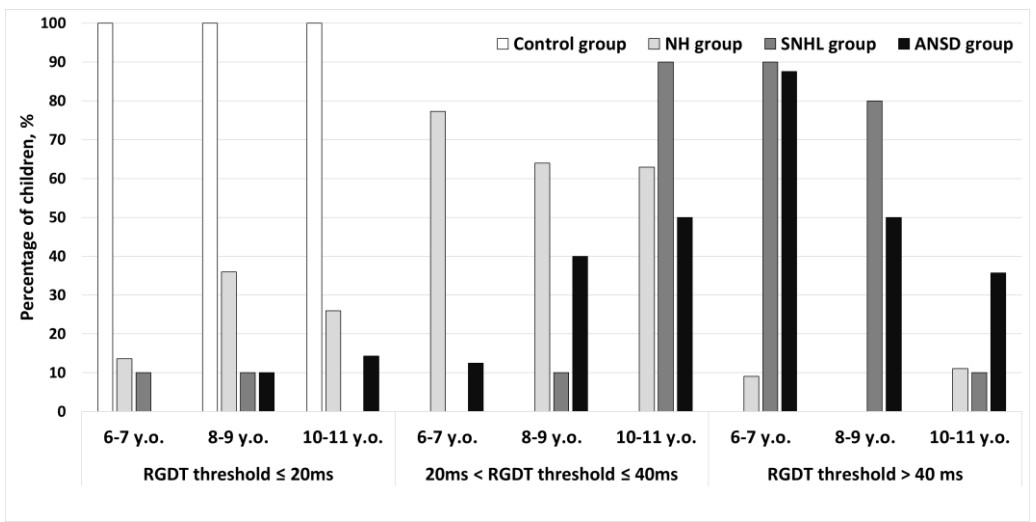

**Figure 1.** The Random Gap Detection Test performance in preterm and full-term children. Control—group of full-term children with normal hearing; NH—group of preterm children with normal hearing; SNHL—group of preterm children with sensorineural hearing loss; ANSD—group of preterm children with auditory neuropathy spectrum disorder.

The mean gap detection thresholds for normal-hearing preterm and full-term children who successfully completed the test are presented in Figure 2. All age subgroups of preterm children with normal hearing showed significantly lower test results compared with their peers from control subgroups ($p < 0.05$ for all age subgroups). A significant improvement in gap detection threshold with age was observed in the control full-term group. A significant difference was observed between the 6–7 and 8–7-year-old subgroups ($p < 0.01$), as well as between the 6–7 and 10–11-year-old subgroups ($p < 0.01$). Preterm children with normal hearing showed no age-related improvement.

Results from the ANSD and SNHL groups showed significant variability in individual values of the gap detection threshold among hearing-impaired children. In the 6–7-year-old subgroup, only one child with SNHL and one child with ANSD successfully completed RGDT; in the subgroup of 8–9 year-olds, only five children with ANSD and two children with SNHL successfully performed the test. One child with SNHL from the subgroup of 6–7 year-olds and two children (one with SNHL and one with ANSD) from the subgroup of 8–9 year-olds demonstrated normal gap detection thresholds (two of them had mild hearing loss and one had a moderate degree of hearing loss).

All preterm children demonstrated significantly lower performance compared with the control group ($p < 0.01$). The mean gap detection thresholds of 10–11-year-old children with hearing loss were equal to 32.1 ± 5.6 ms and 36.1 ± 8.8 ms for the SNHL group and ANSD group, respectively. Preterm children from the SNHL and NH groups showed no significant difference. Children with ANSD performed significantly worse ($p < 0.01$) than their preterm peers with normal peripheral hearing.

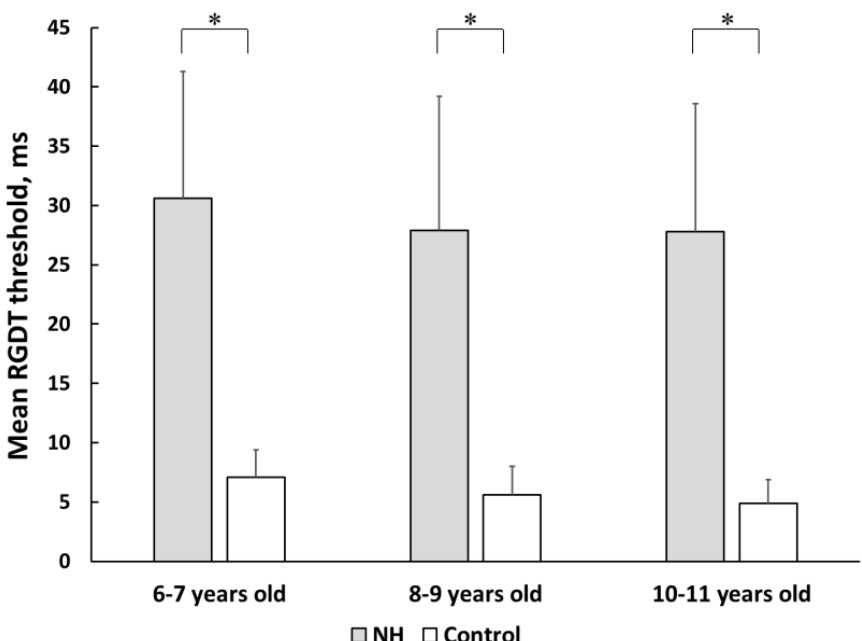

**Figure 2.** The mean gap detection thresholds in preterm and full-term children with normal peripheral hearing. NH—group of preterm children with normal hearing; Control—group of full-term children with normal hearing. Error bars indicate standard errors of the mean. Asterisks denote statistical significance: * $p < 0.05$.

### 3.2.2. Duration Pattern Sequence Test

Figure 3 displays the results of the Duration Pattern Sequence (DPS) test in three age subgroups of children born preterm and at term. The mean rate of correct rhythm recognition was 78.7 ± 15% in the control group, 44.1 ± 20% in the NH group, 41 ± 17.6% in the SNHL group, and 40 ± 20.1% in the ANSD group. Within the NH group, two children (3%) faced difficulties; in the SNHL group, only one child (3%) struggled with the test and in the ANSD group, four children (11%) did not complete the test successfully.

Preterm children with normal hearing demonstrated significantly poorer performance in the DPS test compared with the control group in all age subgroups ($p < 0.01$). In the 8–9-year-old subgroup of NH preterm children, the DPS test results were comparable to those obtained in hearing-impaired children; however, the 6–7-year-old subgroup of NH children demonstrated significantly poorer performance than their hearing-impaired peers ($p < 0.01$). In these children, the test results improved as they got older, and a significant improvement was found at the age of 8–9 years ($p < 0.01$).

Preterm children with hearing impairment also demonstrated significantly poorer performance on the DPS test compared with the control group ($p < 0.01$). No significant differences were found within the 6–7 and 8–9-year-old subgroups of these children. However, at the age of 10–11 years, preterm children with ANSD demonstrated significantly worse test results compared with their preterm peers with normal hearing ($p < 0.01$). No significant difference was found between the 10–11-year-old children with SNHL and their peers from the NH group. A significant improvement in rhythm recognition was observed in 10–11-year-old children with SNHL compared with the youngest 6–7-year-old subgroup ($p < 0.01$). Children with ANSD showed no improvement in test performance with age.

In the control group of healthy full-term children, an improvement in test results with age was found: a significant difference was observed between the 6–7 and 10–11-year-old subgroups ($p < 0.01$), as well as between the 8–9 and 10–11-year-old subgroups ($p < 0.01$).

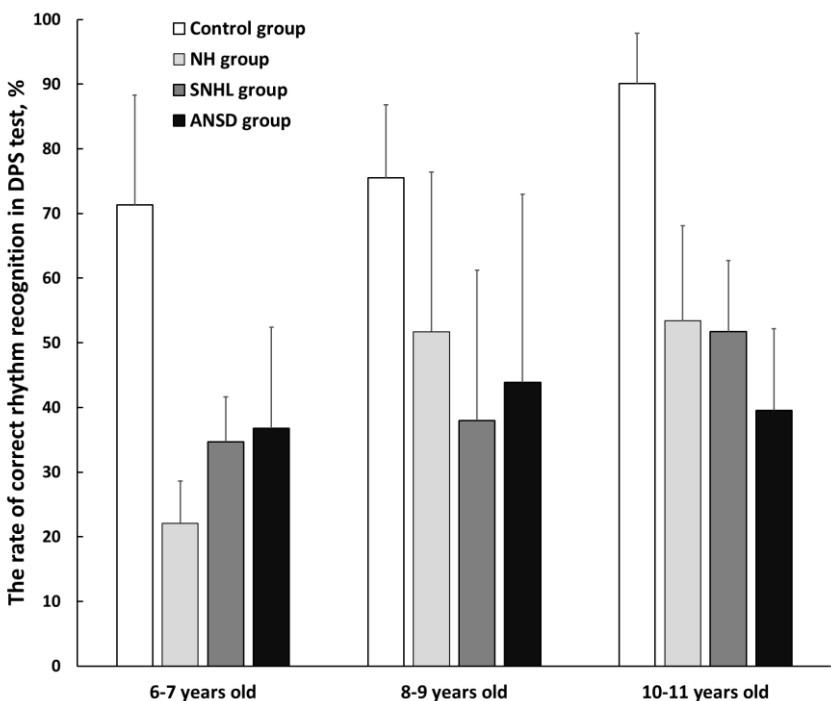

**Figure 3.** The mean rates of correct rhythm recognition in three subgroups of preterm and full-term children. NH—group of preterm children with normal hearing; SNHL—group of preterm children with sensorineural hearing loss; ANSD—group of preterm children with auditory neuropathy spectrum disorder; Control—group of full-term children with normal hearing. Error bars indicate standard errors of the mean.

### 3.2.3. Assessment of the Intelligibility of Sentence Speech in Quiet

The results of the Simplified RuMatrix test in quiet for children 6–7 and 8–9 years old are presented in Table 3. The speech recognition thresholds in quiet (SRTQ) in normal-hearing preterm children did not differ significantly from the results for full-term children. The NH and control groups showed a significant improvement in test performance with age ($p < 0.05$).

**Table 3.** Results of the Simplified RuMatrix and RuMatrix tests in quiet in 6–7, 8–9- and 10–11-year-old children.

| | NH Group | SNHL Group | ANSD Group | Control Group |
|---|---|---|---|---|
| | *Simplified RuMatrix test in quiet (SRT_Q, dB SPL)* | | | |
| 6–7 years old | $26.6 \pm 7.1$ | $40.5 \pm 4$ | $41.4 \pm 5$ | $23.0 \pm 4.1$ |
| 8–9 years old | $20.9 \pm 5.4$ | $34.1 \pm 12$ | $39.4 \pm 3.9$ | $20.2 \pm 3.4$ |
| | *RuMatrix test in quiet (SRT_Q, dB SPL)* | | | |
| 10–11 years old | $26.1 \pm 7.6$ | $29.6 \pm 7$ | $38.4 \pm 7.4$ | $19.4 \pm 1.6$ |

Results are reported as mean $SRT_Q \pm SE$, dB SPL.

The mean $SRT_Q$ in subgroups 6–7 and 8–9 years old was significantly higher (worse) in hearing-impaired preterm children compared with NH preterm ($p < 0.01$) and full-term children ($p < 0.01$). The performance of children in the SNHL and ANSD groups did not have a significant improvement in the Simplified RuMatrix test in quiet with age.

The results of the RuMatrix test for children aged 10–11 years old (Table 3) indicated that the SRTQ in the NH preterm group was significantly higher compared with the control full-term group ($p < 0.05$).

The mean SRTQ in hearing-impaired preterm children aged 10–11 years old was significantly poorer compared with the control full-term group ($p < 0.01$). Moreover, a significant difference in SRTQ was observed between the ANSD and SNHL groups ($p < 0.05$), with ANSD children demonstrating the lowest performance. Furthermore, children with ANSD performed significantly worse when compared with their preterm peers with normal hearing ($p < 0.01$).

### 3.2.4. Assessment of the Intelligibility of Sentence Speech in Noise

The results of the Simplified RuMatrix test in noise for children 6–9 and 8–9 years old are presented in Figure 4A. Preterm normal-hearing children showed good results on the test, and their results did not differ significantly from the results of the control group. The children aged 6–9 years old showed age-related improvement in sentence intelligibility in noise ($p < 0.01$).

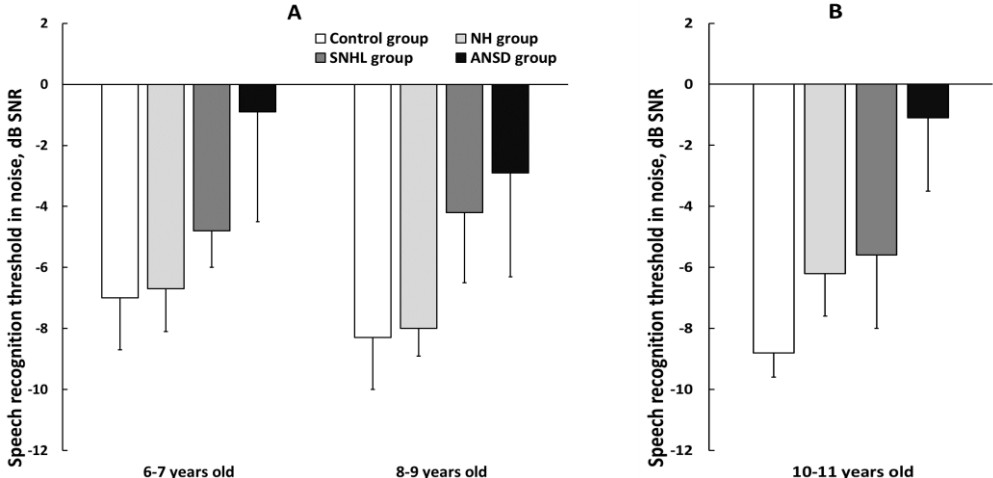

**Figure 4.** The mean speech recognition threshold in noise ($SRT_N$) in three subgroups of preterm and full-term children. (**A**) Results of the Simplified RuMatrix test in 6–7 and 8–9-year-old children; (**B**) results of the RuMatrix test in 10–11-year-old children. NH—group of preterm children with normal hearing; SNHL—group of preterm children with sensorineural hearing loss; ANSD—group of preterm children with auditory neuropathy spectrum disorder; Control—group of full-term children with normal hearing. Error bars indicate standard errors of the mean.

A significant difference in speech recognition threshold in noise ($SRT_N$) was observed between ANSD and NH groups ($p < 0.01$) and between ANSD and control groups ($p < 0.01$), with ANSD children demonstrating the worst test performance. Additionally, children with SNHL exhibited notably poorer comprehension of speech in noise compared with the control group ($p < 0.01$).

The $SRT_N$ for 10–11-year-old children evaluated using the RuMatrix test is presented in Figure 4B. The results revealed that preterm children with normal hearing showed significantly lower speech intelligibility when compared with the control full-term group ($p < 0.01$).

The speech intelligibility in hearing-impaired preterm children was lower when compared with the control full-term group ($p < 0.01$). The preterm ANSD group demonstrated significantly worse performance when compared with their preterm peers from the SNHL ($p < 0.01$) and NH groups ($p < 0.01$). Three children with ANSD did not pass the test. The preterm SNHL and NH groups showed no difference in speech recognition threshold in noise.

## 4. Discussion

Due to the complex anatomy of the central auditory system, auditory processing cannot be evaluated using only one method. Auditory processing assessment requires the use of a set of specialized tests. Traditionally, four main categories of psychoacoustic tests are most widely used in clinical practice for the evaluation of central auditory processes: (1) monaural low-redundancy speech tests; (2) dichotic listening tests; (3) measures of binaural interaction and (4) temporal processing tests [20]. However, when examining hearing-impaired patients, who are hearing aid or cochlear implant users, the choice of tests becomes very limited; in particular, it is impossible to evaluate binaural interaction and dichotic testing. Taking into account the possibility of auditory processing assessment in the conditions of using hearing devices, the temporal processing tests, as well as speech audiometry in quiet and noise in the free sound field were selected for this study.

### 4.1. Temporal Processing Evaluation

The present study focuses on evaluating two components of auditory temporal processing: temporal resolution and temporal ordering. It is assumed that the temporal processing in the CANS is primarily based on the bilateral brain hemisphere's function and the auditory cortex. Additionally, the state of the brainstem, involved in primary analysis and effective conduction of acoustic input through complex cross-links, and the corpus callosum, providing interhemispheric integration, are the key components crucial for this high-order auditory skills [20,57].

Our study revealed that in tests evaluating temporal resolution (RGDT) and the ability to recognize rhythm (DSP test), all premature children of all age subgroups, both normal hearing and hearing impaired, showed worse results than the control group. This is consistent with the results of previous studies, which found that the violation of the temporal processing of sound information is most characteristic of children born prematurely [32–34].

### 4.1.1. Temporal Resolution Testing

Previous studies have established that normally in full-term children, by the age of 7 years, RGDT values (for tonal stimuli) remain stable and reach the values recorded in adults [51,58,59]. This is due to the fact that the temporal resolution ability primarily depends on the state of the brainstem, which normally becomes fully developed by the age of 4–5 years [20,60,61]. At the same time, both hemispheres of the brain and the auditory cortex, which undergo continuous progressive maturation during childhood, are also involved in the development of this central auditory function [20]. However, the results of the control group in the current study demonstrated that a reduction in the gap detection threshold can persist until the age of 8–9 years. This phenomenon is likely linked to the continued maturation of the brainstem structures that undergo changes in myelination and/or synaptic density until at least the end of adolescence. The improvement in RGDT results with age can be attributed to the ongoing maturation during childhood [62–65]. The absence of age-related RGDT improvements in premature children with normal hearing may be associated with a delay in the development of the brainstem and auditory cortex.

Preterm children from the SNHL and NH groups showed no significant difference, while children with ANSD performed RGDT significantly worse than their normal-hearing preterm peers even at 10–11 years of age. Impairments in temporal resolution processing in children with ANSD obtained in the present study, were also reported in children with no history of preterm birth [66]. ANSD, at least at the initial stages of the pathological process, is characterized by intact function of the outer hair cells but impaired functioning of the auditory nerve that may be the result of pathophysiological mechanism activation, such as deafferentation (due to the afferent impulses interruption or decrease in numbers of activated type I nerve fibers) and desynchronization (due to suppression of an action potential) processes. Disrupted neural activity significantly impairs temporal processing, which is manifested by the deterioration of RGDT results [67–69].

#### 4.1.2. Temporal Ordering Testing

The temporal ordering ability, as a higher-order function of central auditory processing, requires appropriate maturity of various levels of the auditory pathway, including the brainstem and the auditory cortex, with the corpus callosum providing the bilateral brain hemisphere's functioning [70–72]. Previous research data have shown that normally this ability matures by the age of 11–12 years [71], which is consistent with the results obtained in the present study. Specifically, a significant improvement in rhythm recognition in healthy full-term children was found between the ages of 8 and 11 years. Similar results were reported in other studies [71,73]. Moreover, our results revealed that normal-hearing preterm children showed worse performance in the DSP test compared with full-term children. The rhythm recognition performance improved with age but did not reach the values registered for healthy control children. In normal-hearing preterm children, unlike the children of the control group, an improvement in test performance was revealed by the age of 8–9 years, while age-related changes were insignificant in the older children aged 10–11 years. Similar results were reported by Durante et al. (2018) in preterm children aged 8–10 years [32]. This may be due to combined abnormalities in brain structures, particularly the delayed/abnormal development of the corpus callosum [74,75] and temporal cortex areas [18]. These dysfunctions can persist into adolescence and even into early adulthood [76]. The results suggest that the observed dysfunction may be partially alleviated due to progressive maturation and adaptive neuroplasticity effects as normal-hearing preterm children mature.

Hearing-impaired preterm children demonstrated the worst performance on the DSP test compared with the control group. The younger and middle-aged subgroups with ANSD and SNHL showed no differences in rhythm recognition; however, by the age of 10–11 years, children with ANSD showed dramatically worse results compared with SNHL. In addition, complex structures of the central auditory system, critical for the high-order components of temporal processing, may be severely affected in ANSD, which may be particularly manifested in the morphofunctional abnormalities of the corpus callosum and auditory cortex [77–79]. On the other hand, children with SNHL (mainly due to damage to outer hair cells) demonstrated no significant deficit in temporal ordering ability. DSP test results in these children were better and comparable to those of preterm peers with normal hearing. Similar results were reported by Koravand et al. (2010) in children aged 8–13 years, although the authors did not specify a history of preterm birth in the sample [40].

#### 4.2. Speech Audiometry in Quiet

Speech intelligibility is an integrative index of the central auditory system function. The results of the speech intelligibility test in quiet in healthy full-term children revealed a significant improvement in speech recognition threshold with age, indicating ongoing maturation processes in the auditory system. The younger and middle-aged subgroups of normal-hearing preterm and full-term children displayed similar results in speech recognition threshold in quiet. However, by the age of 10–11 years, significant differences were found, which was likely due to the use of more complex speech material in the RuMatrix test.

The results revealed that preterm children with hearing loss aged 6–9 years had significantly poorer speech recognition in quiet compared with both normal-hearing preterm and full-term peers. The possible explanation for this finding is the difference in pure-tone thresholds, which, even with adequately fitted hearing aids, are 10–20 dB lower compared with normal pure-tone thresholds. The high correlation between $SRT_Q$ and outcomes of pure-tone audiometry was reported in previous studies [80]. Our findings confirm the existing understanding of the interrelationship between temporal processing and speech intelligibility abilities [67,81,82], which was clearly demonstrated in hearing-impaired preterm children in the present study.

### 4.3. Speech Audiometry in Noise

Speech intelligibility in noise, as a higher-order auditory processing, is primarily related to the auditory cortex functioning. In our study, the healthy full-term children demonstrated a positive trend in $SRT_N$ with age, and similar results were observed in the normal-hearing preterm group, which is likely due to active maturation and adaptive neuroplasticity in the developing brain. Children with SNHL had significantly poorer $SRT_N$ compared with the control group, which may be explained by a combined effect of hearing loss and prematurity negatively affecting speech recognition in the presence of competing stimuli [34,83]. The preterm group with ANSD showed no significant changes in $SRT_N$ with age. Furthermore, similarly to the results of the entire test battery, children with ANSD demonstrated dramatically poor speech intelligibility in noise. This may be due to the significant impairment of temporal processing in ANSD that adversely affects speech recognition, especially in noisy environments [68,69,84]. Moreover, subgroups of 6–7 and 8–9-year-old children with SNHL and ANSD showed no improvement in $SRT_N$. However, at an older age, the SNHL group demonstrated no difference with their normal-hearing preterm peers, possibly indicating positive dynamics in speech processing by early adolescence and hearing deficit compensation (due to appropriate hearing aid fitting).

Despite the observed improvement in $SRT_N$ across all groups with age, the oldest subgroup of preterm children, both with and without hearing loss, had significantly poorer $SRT_N$ compared with their healthy full-term peers, with the ANSD subgroup demonstrating the worst results. These findings can be explained by delayed and/or disrupted development of the central auditory system in preterm children who are not fully compensated by the age of 10–11 years. Notably, hearing-deficit compensation in children with ANSD may be achieved through cochlear implantation. Applying an electrode array for direct stimulation of spiral ganglion cells can positively improve the auditory nerve activity impaired by peripheral hearing defects [85]. The significant improvement in speech intelligibility, even in complex acoustic environments, was largely reported in implanted patients [84,86]. However, only three children with cochlear implants participated in our study, so it was not possible to evaluate the effect of this type of hearing device. The investigation of central auditory processing in preterm cochlear implant users may be the goal of future studies.

Thus, the results of the present study indicate that children born preterm exhibit varying degrees of central auditory system dysfunction, likely reflecting a generalized morphofunctional deficit in auditory processing. This is consistent with the limited research data available in the literature [66,83]. The auditory processing deficit in preterm children found in the present study is exacerbated by the presence of peripheral hearing impairments that are more pronounced in the case of ANSD.

Moreover, the results of the present study confirm the urgency for a detailed comprehensive audiological assessment of both hearing-impaired and normal-hearing preterm children in order to identify possible signs of APD. Analyzing and understanding the auditory processing specificity in children born preterm can positively contribute to more effective implementation of rehabilitation programs in this population. The first-choice strategy in the rehabilitation process is the auditory training program [20]. Improvements in central auditory processing skills in children after auditory training have been proven through both psychoacoustic and electrophysiological testing [87,88].

### 4.4. Study Limitations

Firstly, due to the relatively small sample size of hearing-impaired children, it was not possible to reliably evaluate the effect of perinatal risk factors and comorbid pathology on the central auditory system status. Further research is needed.

Secondly, in the present study, we analyzed the results obtained in different acoustic conditions: free sound field testing for hearing-impaired children and headphone testing for children with normal peripheral hearing. Typically, such results are difficult to compare. However, normally, speech test results in free sound field conditions prove to be better than monaural headphone testing ones [89], which may be due to the advantages of binaural

hearing over monaural hearing. This fact allowed us to analyze the results obtained in different acoustic conditions, as even children with normal hearing tested with headphones performed better than hearing-impaired children tested in free sound field conditions.

## 5. Conclusions

Children born preterm, even with intact peripheral auditory function, can exhibit signs of APD, which may be partially eliminated with age. A similar positive dynamic is found in hearing-impaired preterm children with SNHL, whereas hearing-impaired children with ANSD born preterm may experience persistent deficits in central auditory processing abilities until early adolescence.

The central auditory system dysfunction in preterm children appears to be multilevel in nature, but the greatest deficit, which is not compensated with age, is observed in the auditory temporal processing abilities.

Prematurely-born children require ongoing audiological monitoring because of the risk of developing APD, particularly if they exhibit delayed speech and language development.

**Author Contributions:** Conceptualization, I.V.S. and M.Y.B.; methodology, I.V.S. and E.S.G.; software, E.S.G.; validation, I.V.S., E.S.G., M.J.V. and V.M.K.; formal analysis, I.V.S. and E.S.G.; investigation, I.V.S., E.S.G., M.J.V. and V.M.K.; resources, M.Y.B.; data curation, I.V.S.; writing—original draft preparation, E.S.G., M.J.V. and V.M.K.; writing—review and editing, M.J.V. and V.M.K.; visualization, E.S.G., I.V.S. and V.M.K.; supervision, M.Y.B. and M.J.V.; project administration, M.J.V.; funding acquisition, M.J.V. All authors have read and agreed to the published version of the manuscript.

**Funding:** This research was funded by the Russian Science Foundation, grant number 23-25-00108.

**Institutional Review Board Statement:** The study was conducted in accordance with the Declaration of Helsinki, and approved by the Ethics Committee of St. Petersburg Psychological Society (protocol #21, 6 April 2023).

**Informed Consent Statement:** Informed consent was obtained from all the subjects/subjects' parents involved in the study.

**Data Availability Statement:** The data presented in this study are available on request from the corresponding authors. The data are not publicly available due to our intention to know the research groups interested in our study and data, and about the methods and research they are engaged in.

**Conflicts of Interest:** The authors declare no conflict of interest.

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
