# Peer review of "Impact of Prematurity on Auditory Processing in Children"

_pathophysiology, doi:10.3390/pathophysiology30040038_

Round 1

Reviewer 1 Report

Comments and Suggestions for Authors

This interesting paper requires more work needed to present an unfinished paper.  The paper focuses on 6–11-year-old hearing impaired that shows impaired preterm children.  The population is small (241 children overall with 132 premature born children) that makes it impossible to evaluate the effect of perinatal risk factors. Moreover, I suggest presenting the cohorts in a table.  The population is very small (1-29 group 1, 2-21 group 2, ?? of group 3) with a large variation.  

Several set of data claim to show significant differences but with large range of overlap combined with a low number of children remains unclear to me.

Introduction

P32 Expand premature children in the world.

P37 Causality of SNHL that leads to ANSD? These are probabilities, not proven connections from p35-40.

P59 suggesting a citation (Molnar et al.. 2020; Yamoha et al., 2023)

P82 We know that children can hair prior to birth and can speak at about 2 years of age (see Yamoah et al., 2023, for details

Material and Methods:

The population differs in three groups (32 bilateral neuropathy, 30 bilateral SNHL and 74 preterm children with normal hearing.

P143 Size of each cohort is small.  Provide a table to illustrate the preterm and normal born children.

P249 Fig. 1 Suggest presenting the age group in ANSD, SNHL, NH and control at age 6-7.8-9 and 10-11.

P262 Fig. 2 move this information to supplement.  A significance is needed in the figure.  The error bar is very large.  How did they test for p??

Table 1 is fine but is closely identical to Fig.2.  what was the size of children??

Table 2 I want to have the numbers of children for each group of hearing loss per age group.

Fig 4 can combine the three groups of hearing loss per age group.

The discussion needs to be shortened from 9-14 to about 9-12.  Please keep in mind that I will request in certain areas a short expansion.

p420-432I suggest rewriting this chapter.  More information is provided in Molnar et al., 2020 and Yamoah et al., 2023 that also cited several other papers to highlight the infant development.

P451-477 Further details of the genetics of auditory nuclei is presented in Elliott et al., 2021, 2022.  Keep in mind that the entire development depends on Lmx1a/b DKO in mice (see Chizhikov et al., 2021).  Several cases are described after LMX1A in humans.  I suggest providing several more recent reviews that detail the effect on auditory nuclei.

P478-493 I suggest citing the work of Molnar et al., 2020 that details the human (and mouse) early development.  I also cite and work into the revised version of Mukherjee & Kanold, 2023. Keep in mind that work of Tomblin et al (1997) defined specific language impairment in kindergarten children, long before the current assessment started at 6 old infants that is affecting the handedness (see also Abbondanza et al., 2022).

Comments on the Quality of English Language

Language can be improved.

Reviewer 2 Report

Comments and Suggestions for Authors

1. In the introduction please add the infromation about the possble pharmacological treatment option of the patients who suffer from SSNHL and cite:

1. Skarżyńska MB, Kołodziejak A, Gos E, Sanfis MD, Skarżyński PH. Effectiveness of Various Treatments for Sudden Sensorineural Hearing Loss-A Retrospective Study. Life (Basel). 2022 Jan 10;12(1):96. doi: 10.3390/life12010096. PMID: 35054488; PMCID: PMC8779405.

and 

2. Ren H, Hu B, Jiang G. Advancements in prevention and intervention of sensorineural hearing loss. Ther Adv Chronic Dis. 2022 Jun 27;13:20406223221104987. doi: 10.1177/20406223221104987. PMID: 35782345; PMCID: PMC9243368.

It is very important to add the information about the limitations of the results of your study. It will give a chance for the potential reader to have an information what have to be still examined. 

no more comments now

Reviewer 3 Report

Comments and Suggestions for Authors

The present study aimed to evaluate auditory processing in three groups of 6-11-year-old hearing impaired, with ANSD (Group 1) and SNHL (Group 2) and normal hearing preterm (Group 3) and healthy normal hearing full-term children. The 3 preterm Groups were divided in age intervals – 6-7, 8-9 and 10-11 years. The children are subjected to adequate audiological testing in order to assess peripheral and central auditory processing. The study revealed that both groups of hearing impaired and group of normal hearing preterm children scored lower on the entire test battery compared to the control group of healthy full-term children. There were significant differences found in auditory processing abilities between three groups of preterm children, partially eliminated by age. Positive dynamic changes are found in hearing impaired preterm children with SNHL, whereas hearing impaired children with ANSD born preterm may experience persistent deficits in central auditory processing abilities.

This is a very good study on an important subject – auditory processing in preterm children. The study is well designed and performed. Though the number of children in some of the age groups are few the results seem reliable. It is a massive text with numerous references. I am ready to recommend this manuscript for publication in Pathophysiology but before that the authors should respond to the following comments:

Both Introduction and Discussion are written in a very extensive text. Would it be possible to somewhat reduce the lengths of these sections to easify for the reader?

Had it been easier to follow the text if the various groups, subgroups had been clearly defined as 1a, 1b, 1c, 2a.. etc and these group names been consequently used in the text?

The text of the first section of the Results concerning The peripheral hearing assessment had been easier to read in a table. Furthermore I would have liked to see the various degrees of hearing loss  – mild/moderate/moderately severe/moderate-moderately severe/severe etc –defined in a dB-scale.

Reviewer 4 Report

Comments and Suggestions for Authors

In their manuscript "Auditory Processing in Preterm Children with Peripheral Sensorineural Hearing Loss" Maria Boboshko et al. describe measures of auditory processing in preterm children with HI due to SNHL and ANSD as well as in  normal hearing preterm children.   SNHL and NH groups showed age-related improvement in speech recognition thresholds in noise. However, children with ANSD at the age of 10-11y showed no age-related effect and higher SRTN. Overall, peripheral hearing loss, even with the use of appropriate hearing devices, negatively affected auditory processing in preterm children. Children with ANSD performed worse. Children born preterm with intact peripheral auditory function show signs of APD, that may be partially eliminated with age.

Overall, the manuscript in its current form is hard to read, introduction and discussion are far to long and a red line is missing. However, the study shows several strengths (comparrision of SNHL, ANSD and normal heraing preterms, extensive audiological testing, comparrison of age groups etc.) and the findings and implications are relevant and important. From the view point of the reviewer these data should be published after major revision:

1) Titel: Should be changed as it is a bit missleading by only mentioning children with peripheral SNHL.

2) Introduction: Has to be shortened and restructered. A red line is missing.

3) Materials and Methods: - How was ANSD diagnosed in those children? - Hearing aids: information on time of fitting? Any family centered early intervention? - The authors state that preterm children with a "sufficient level of speech develooment" were included, however, it remains unclear how this sufficient level was determined or tested. - 

4) Results: Again, too expansive in the description. Could be more to the point and less narrative. Given figures are appropriate and very helpfull.

5) Discussion: Far to long. Red line is missing. It should be discussed to which extent the used test battery is developmentally appropriate. How was the appropriateness of the test batteries for the included children determined?

6) Conclusion: What are the practical implications of these findings? Implications on HL testing in these children and the interpretation of the results. Implication for early intervention. Implications for fitting of hearing devices. Implication for follow up?

Comments on the Quality of English Language

Quality seems ok but moderate editing is required.

Reviewer 5 Report

Comments and Suggestions for Authors

Please see file attached. Thank you  for the opportunity  to  read your work . I am sorry if asked more explanations. I think that this article has  good potentials

Regards

Round 2

Reviewer 1 Report

Comments and Suggestions for Authors

This paper is improved but has not cited relevant papers.  For example, Molnar et al is now cited but does not cite the work of Yamoah et al., 2023.  Likewise, certain aspects are mentioned in the review that does not cite the work of Elliott et al., 2022, for example.  Stating that the papers are included but do not cite the relevant papers.

Comments on the Quality of English Language

English is fine

Reviewer 4 Report

Comments and Suggestions for Authors

After Major revision all questions and suggestions of the reviewer are answerd. No further comments. From my point of view paper is ready for acceptance.

Comments on the Quality of English Language

 Minor editing of English language required

Reviewer 5 Report

Comments and Suggestions for Authors
